A 3D-printed passive exoskeleton for upper limb assistance in children with motor disorders: proof of concept through an electromyography-based assessment

Sanchez Cristina cristina.sanchezlopezpablo@ceu.es
Blanco Laura
del Río Carmina
Urendes Eloy
Costa Vanina
Raya Rafael
Departamento de Tecnologías de la Información, Escuela Politécnica Superior, Universidad San Pablo-CEU, CEU Universities , Urbanización Montepríncipe , Madrid , Spain
Greco Gianpiero
Electronic publication date: 2023 Mar 29
Publication date: 2023
Volume: 11
Electronic Location ID: e15095
Received 2022 Sep 20; Accepted 2023 Feb 28
Copyright: ©2023 Sanchez et al.
Copyright year: 2023
Copyright holder: Sanchez et al.
License: This is an open access article distributed under the terms of the Creative Commons Attribution License, which permits unrestricted use, distribution, reproduction and adaptation in any medium and for any purpose provided that it is properly attributed. For attribution, the original author(s), title, publication source (PeerJ) and either DOI or URL of the article must be cited.
License URL: https://creativecommons.org/licenses/by/4.0/

Keywords: 3D printing, Children, Electromyography, Motor disorders, Passive exoskeleton, Upper limb

Funding: MCIN/ AEI/ 10.13039/501100011033/ FEDER, UE RTI2018-097122-A-I00 PID2021-127096OB-I00 This research was funded by MCIN/ AEI/ 10.13039/501100011033/ FEDER, UE, grant numbers RTI2018-097122-A-I00 and PID2021-127096OB-I00. The funders had no role in study design, data collection and analysis, decision to publish, or preparation of the manuscript.

==============================
The rehabilitation of children with motor disorders is mainly focused on physical interventions. Numerous studies have demonstrated the benefits of upper function using robotic exoskeletons. However, there is still a gap between research and clinical practice, owing to the cost and complexity of these devices. This study presents a proof of concept of a 3D-printed exoskeleton for the upper limb, following a design that replicates the main characteristics of other effective exoskeletons described in the literature. 3D printing enables rapid prototyping, low cost, and easy adjustment to the patient anthropometry. The 3D-printed exoskeleton, called POWERUP, assists the user’s movement by reducing the effect of gravity, thereby allowing them to perform upper limb exercises. To validate the design, this study performed an electromyography-based assessment of the assistive performance of POWERUP, focusing on the muscular response of both the biceps and triceps during elbow flexion–extension movements in 11 healthy children. The Muscle Activity Distribution (MAD) is the proposed metric for the assessment. The results show that (1) the exoskeleton correctly assists elbow flexion, and (2) the proposed metric easily identifies the exoskeleton configuration: statistically significant differences (p-value = 2.26 ⋅ 10−7 < 0.001) and a large effect size (Cohen’s d = 3.78 > 0.8) in the mean MAD value were identified for both the biceps and triceps when comparing the transparent mode (no assistance provided) with the assistive mode (anti-gravity effect). Therefore, this metric was proposed as a method for assessing the assistive performance of exoskeletons. Further research is required to determine its usefulness for both the evaluation of selective motor control (SMC) and the impact of robot-assisted therapies.

Introduction

Physical therapy in the rehabilitation of children with motor disabilities mainly focuses on motor intervention. Bimanual training, constraint-induced movement therapy, fitness training, strength training, and task-specific training, among others, have been shown to be effective in enhancing baseline motor, sensory, and perceptual skills, and learning capabilities (Novak et al., 2020).

In this context, the robotic-assisted approaches aim at helping improve the motor control and muscle strength of these patients (Chia Bejarano et al., 2016). An exoskeleton is an assistive and wearable technology that helps people with motor disabilities or impairments to restore, improve, or at least maintain their functional abilities. Although the idea of exoskeleton was dated back to the late 19th century, the first prototype of a successful one (called Hardiman) was not developed until the 1960s and was thought for military purposes (Gull, Bai & Bak, 2020). Years later, Kazerooni (1990) developed an upper-limb exoskeleton exploring the idea of physical human–robot interaction. Later (2003), researchers from the University of Tsukuba presented the exoskeletal robotics suite HAL (Hybrid Assistive Leg) originally developed to help disabled people in ADL (activities of daily living) (Kawamoto et al., 2004). In the last two decades, the use of upper-limb exoskeleton both for services and rehabilitation has gained attention in the biomedical field as potential solutions for physically weak or disabled people (Islam et al., 2017). Devices such as InMotion® (Fasoli et al., 2008), Haptic Master® (Fluet et al., 2009), and Armeo Spring® (El-Shamy, 2018; Cimolin et al , 2019; Obiedat, 2013) have been demonstrated to be effective complements to physical therapy, especially for motor rehabilitation of the upper limbs. Moreover, in recent years, with the rise in 3D design and printing, the development of low-cost exoskeletons inspired by these commercial devices has increased remarkably (Zeiaee et al., 2017; Oguntosin et al., 2017; Tsai, Yang & Chen, 2019).

POWERUP is a 3D-printing-based passive upper-limb exoskeleton (Fernández et al., 2021) designed to assist upper-limb movement in children with motor disabilities. The device could be used not only with rehabilitation purposes, but also as a re-educational path in children with different temporary, progressive, or permanent physical conditions implying postural and balance deficits as well as difficulties in the movement and/or coordination of the upper limbs such as cerebral palsy (CP) (Fonvig et al., 2021), juvenile arthritis (Patti et al., 2017), spina bifida (Jewell et al., 2010) or muscular dystrophy (Jeannet et al., 2011), among others.

The design of POWERUP is partially inspired by the Wilmington Robotic Exoskeleton (WREX) (Rahman et al., 2006; Rahman et al., 2000; Sanchez et al., 2006), which is a 4 degrees-of-freedom (DoF) mechanism with two rotations at the shoulder and two rotations at the elbow that passively counterbalances the weight of the arm using elastic bands (Zariffa et al., 2012). Moreover, POWERUP adds an extra DoF that allows pronosupination of the elbow.

The assessment and validation of the assistive performance of the exoskeleton are key points before evaluating it with patients in clinical trials. However, there is a lack of well-established methods and metrics for this purpose in the clinical practice. Nevertheless, in the industrial field, the evaluation criteria and metrics are clear and well-classified (Pesenti et al., 2021).

Therefore, in this area, measurement of muscle activity (intensity of muscle contraction) using surface electromyography (sEMG) is the most common technique for evaluating exoskeleton assistance (Pesenti et al., 2021). The greater the amount of assistance provided to a muscle, the lower the level of muscle activation (Pesenti et al., 2021; Coscia et al., 2014; Scano et al., 2015). To quantify this effect, it is important to compare the sEMG registers of users conducting different tasks with and without the assistance of the exoskeleton. The root mean square (RMS) of the sEMG signal, calculated using a moving window from the raw register, is considered to provide the most insight on the amplitude of the EMG since it gives a measure of the power of the signal. In fact, the peak value of the RMS in dynamic tasks, as well as its time-average value in static ones, are metrics commonly used to carry out the aforementioned comparison (Pesenti et al., 2021; Blanco et al., 2019; Huysamen et al., 2018).

Several studies have extrapolated this method of validating assistive performance to assess robotic devices designed for use in the clinical field. Wang et al. (2020) evaluated their assistive system for upper limb motion by comparing muscle fatigue (by means of the muscle activation levels) in three healthy subjects lifting and holding a 1 kg object, both with and without the exoskeleton. Xiao et al. (2018) assessed the assistance of their cable-driven exoskeleton using the RMS values obtained from the sEMG recordings of a series of muscles associated with upper-limb movement in six healthy subjects. Wu et al. (2018) validated the different degrees of assistance of an admittance-based patient-active control upper-limb exoskeleton by comparing the RMS signals of a series of sEMG recordings in three healthy volunteers.

On the other hand, it is important to highlight that the sEMG is a very well-known and commonly used technique in the diagnosis and in monitoring the evolution of patients suffering from different motor impairments. In fact, sEMG has become an important technique for both analyzing the movements and assessing the motor function impairment of children with motor disabilities, as it provides crucial information regarding muscle coordination (Schmidt-Rohlfing et al., 2006; Wei et al., 2019; Bauer, Mall & Jung , 2018; Raouafi, Raison & Sofiane, 2019). Owing to sEMG, information regarding muscle activation, myoelectric manifestation of muscle fatigue, and recruitment of motor units can be obtained (Garcia & Vieira, 2011).

Considering that the sEMG is not only a common technique to evaluate the performance of exoskeletons but also a very well-known technique for diagnosis and follow-up of patients in the clinical field, we hypothesized that:

• The EMG analysis will make it possible to discriminate between the transparent mode (no effect) and the assistive mode (anti-gravity effect) of the exoskeleton.

• The differences between the two modes of performance of the device will lead us to describe if the exoskeleton correctly assists elbow flexion according to the scientific evidence (as stated before, the assistance provided to a muscle should result in lower levels of muscle activation).

• These results will make it possible to postulate this method as a suitable candidate for the validation of this type of robotic device.

Therefore, this study aimed to validate the assistance effect of a 3D-printed passive exoskeleton for elbow flexion by performing an electromyography-based assessment in healthy children that makes it possible to easily distinguish between the transparent and the assistive mode of the device.

This study focused on elbow flexion-extension movements in healthy children, thus laying the foundations and establishing normative reference values for subsequent validation in children with motor disabilities.

Materials & Methods

Upper limb exoskeleton

The POWERUP exoskeleton (Fig. 1) is a five DoF upper limb orthosis without electromechanical actuators and is fabricated using 3D-printing technology. During the design process, a series of expert pediatric physiotherapists from the Instituto de Rehabilitación Funcional La Salle (Madrid, Spain) helped establish the clinical and functional criteria that the device should meet. The present wearable prototype was designed for children aged between 6 and 15 years (arm and forearm lengths between 40 and 60 cm) (Edmond et al., 2020). Its function is to allow elbow and shoulder movements by providing stability as well as anti-gravity weight support when needed, by keeping the wrist and hand in the neutral position.

Figure 1 3D-printed passive upper limb exoskeleton assembly.

Segments: shoulder/back in blue, arm in red, forearm in orange, and hand in green.

POWERUP comprises four structural modules (shoulder/back, arm, forearm, and hand). All of the modules allow the physiotherapist to easily adjust (width, height, and length) the device to adapt to the user’s anthropometry using telescopic bars, anchors, and elastic straps. In addition, this modular design allows for a quick and intuitive assembly. The exoskeleton allows flexion-extension and internal-external rotation of the shoulder, flexion-extension, internal-external rotation, and pronosupination of the elbow. The hand module has a horizontal surface on which the hand rests and the wrist is fastened with straps while resting in a neutral position in the coronal plane. This surface rotates within a concentric sliding ring mechanism, allowing pronation and supination of the elbow.

The device was manufactured by 3D printing (Creality CR-5 Pro® and BQ Witbox 2® printers) with a PLA+ 1.75 mm filament (a thermoplastic monomer derived from renewable and organic sources), which makes it a lightweight, low-cost, and easily reproducible solution. All the pieces were printed with a 0.2 mm layer height, while the rest of the printing parameters were adjusted for each piece. Thus, bar-type parts with holes were printed with the CR-5 printer, with minimum support on the holes, with a 20% fill density, and adding extra fixation to the hot bed owing to the use of spray adhesive and a raft, as some pieces require more than 24 h to be printed. In this case, the parameters according to the printer test were as following: 207 °C with a 60 °C hot bed, 96% flux, and a Bowden extruder with 6–30 mm/s retraction speed. Witbox 2 was used for more complex pieces with circular parts, with 60° supports at a minimum of 25% infill density, and spray adhesive to increase fixation. On this occasion, the parameters according to the printer test were as following: 204 °C, 98% flux, and direct drive with 1–30 mm/s retraction speed. The overall printing time for the entire device was approximately 170 h.

The POWERUP assistive mode provides anti-gravity weight support, helping the children lift and maintain the weight of their arm and enhance elbow flexion movement. The assistance was achieved by placing elastic bands around protruding lugs at the ends of the forearm segments, as depicted in Fig. 2. The use of conventional materials, 3D printing and common elastic bands makes it possible to replicate the exoskeleton easily.

Figure 2 Assistive mode configuration.

Position of the elastic bands to set the assistive mode for the elbow joint.

The POWERUP exoskeleton was anchored to an external metal-rolling frame (Fig. 3). This structure can be easily moved and adjusted according to the user’s shoulder height. Once placed in a convenient position, the wheels were locked to prevent unintentional displacement during training sessions.

Figure 3 POWERUP exoskeleton.

General view of POWERUP exoskeleton on a metallic frame. In this case, two prototypes were assembled for the training sessions with both arms.

Surface electromyography

The measurement of the assistance effect will be assessed on the analyses of muscle activity (thanks to sEMG recordings) of the biceps brachii (agonist in elbow flexion and antagonist in elbow extension) and triceps brachii (agonist in elbow extension and antagonist in elbow flexion).

The root mean square (RMS) signal is proportional to the number of muscle fibers activated in a particular muscle (recruitment) (Dufour et al., 2018; Baggen et al., 2019). On the other hand, the Muscle Activity Distribution (MAD), which is the proposed metric for the POWERUP assessment, is defined as the percentage of muscle activity of a specific muscle over the total amount of muscle activity recorded for a particular movement. This parameter was chosen because (1) it makes it possible to compare the contractile activity between agonist (or synergist) and antagonist muscles and/or to study muscle overactivation/inhibition (Merletti & Farina, 2016); and (2) it can be explored as an alternative for comparison among different subjects instead of the metrics obtained from the normalized RMS, which usually requires the attainment of maximum voluntary contraction (MVC), which can be relatively easy to obtain in healthy children but not in children with motor disabilities for obvious reasons.

The sEMG signals and proposed metrics were obtained using an ultralight wearable device (mDurance™) that integrates a three-dimensional inertial sensor with a two-channel electromyograph. The device is controlled using an Android™ mobile device via Bluetooth. This application enabled the acquisition of two simultaneous sEMG signals with a sampling rate of 1,024 Hz. Each sEMG record was stored in a device cloud service. The recordings were reloaded (and/or downloaded) for subsequent analysis. A cloud service makes it possible to obtain and visualize the time-dependent evolution of the recruitment of muscle fibers in terms of RMS, enabling the observation of the co-activation patterns for the recorded muscles. The MAD can also be visualized using the corresponding bar plots (Fig. 4).

Figure 4 SEMG visualization and metrics.

Example of the visualization of the time-dependent evolution of the recruitment of muscle fibers in terms of the root mean square (RMS, µV) signal, where the graph represents the RMS value associated with the muscular activity of the biceps brachii and triceps brachii (time windows of 0.25 s), and the corresponding bar plots to visualize the muscle activity distribution (MAD, %) (below). This recording represents three elbow flexion-extension movements.

For obtaining the RMS signal, first, a fourth order Butterworth bandpass filter with a cut-off frequency at 20–450 Hz is applied and, second, the resulting signal is smoothed using a window size of 0.25 s. For a specific muscle, the MAD (%) is calculated as the ratio between the result of the time integral of the RMS signal for this muscle over the sum of all the results of the time integrals for each one of the muscles measured in this record (Molina-Molina et al., 2020).

The sEMG recordings were made with solid gel 45 × 42 mm general-purpose disposable Ag/AgCl electrodes that were placed according to the recommendations for sensor locations in arm muscles from the Surface Electromyography for the Non-Invasive Assessment of Muscles (SENIAM) project (http://seniam.org/arm_location.htm), on the line between the acromion and the fossa cubit at 1/3 from the fossa cubit to record biceps brachii activity, and at 50% on the line between the posterior crista of the acromion and the olecranon at two finger widths medial to the line for recording triceps brachii activity (Fig. 5) (Hermens et al., 2000).

Figure 5 Electrode placement.

Electrode placement for the biceps brachii (left) and triceps brachii (right). The yellow crosses represent the location of the electrodes, and the blue dots represent each muscle insertion (http://seniam.org/arm_location.htm). Image credit: Hermens et al. (2000).

Participants

A total of 11 children with normal growth and development (8 males and 3 females, aged 9–10 years old, with average ±standard deviation height, weight, and body mass index (BMI) of 32.0 ± 4.2 kg/m2, 138.9 ± 4.6 cm, and 16.5 ±  1.6, respectively), who were randomly selected from a class of students from the Colegio CEU San Pablo Montepríncipe in Madrid (Spain), participated in the study.

The expected effect size, according to preliminary tests should be large. With a Cohen’s d = 1, the Power Analysis (‘pwr’ library, R statistical computing) resulted in a sample size of 9.9 so that, a sample of 10–11 subjects should be enough to corroborate the hypothesis of this proof of concept.

All children, their parents or legal guardians, and the head office of Colegio San Pablo CEU Montepríncipe provided written informed consent to participate in the study. Ethical approval was obtained from the Research Ethics Committee of the San Pablo CEU University (561/21/53).

Data acquisition protocol

The participants were first informed about the experiment and received clear explanations of the protocol established for the movements they should perform. Each participant sat next to the exoskeleton mounted on the platform. All participants were asked to sit comfortably with their backs straight, trying not to change their trunk position during the experiment. Subsequently, a physiotherapist placed both the sEMG electrodes and the exoskeleton, properly aligned and fixed, on the dominant arm of the child.

Once the wearable electromyography device was properly configured and ready to record the sEMG signals, each child was asked to perform two tests. The first test (Test 1) consisted of recording the sEMG signal of elbow flexion-extension movements when the children wore the exoskeleton, but no elastic bands were added (transparent mode). Test 1 was performed to establish a reference to be compared with the exoskeleton assistance settings. Test 2 consisted of measuring the sEMG signal of elbow flexion-extension movements with the children wearing the exoskeleton configured in its assistive mode. The elastic bands were attached to assist elbow flexion (Fig. 6), thus resisting its extension. No bands were used on the upper arm segment of the exoskeleton in either of the tests to work only with pure elbow flexion-extension movements, keeping the upper arm stable in its position.

Figure 6 Exoskeleton and sEMG configuration.

Exoskeleton and surface electromyography (sEMG) sensors arrangement for elbow flexion assistance (assistive mode).

Each of the two tests (Test 1 and Test 2) started with an sEMG recording of 5 s while the subject remained relaxed and without performing any voluntary movements and/or contractions to check whether the arm was really at rest (no observable increase in basal tone). After the first 5 s, and considering that the fiber recruitment depends on the contraction strength and speed of the exercise, the time evolution of each test was controlled as follows: the subject had to flex the elbow fully (trying to touch the shoulder) and maintain this position for 3 s. Then, the elbow was fully extended and maintained for 3 s. This holding flexion–extension movement was repeated three times for each test. This protocol aims to (1) minimize the muscle fatigue that occurs when an intense effort is maintained over time and (2) favor the ordered recruitment of the motor units (from the smallest to the largest) that generally occurs in controlled movements that do not require intense contraction strength and/or fast performance velocities (Wakeling et al., 2012).

Data analysis

The analysis focused on (1) corroborating the initial hypotheses, demonstrating that the sEMG analysis based on the proposed metric (MAD), discriminates between the transparent and assistive modes of the POWER UP exoskeleton, and (2) studying if the differences in the MAD, comparing Test 1 and Test 2 for both biceps and triceps, are statistically significant. The analysis was conducted using RStudio, an integrated development environment (IDE) for R, which is a programming language for statistical computing and graphics. To determine whether the samples met the assumptions to conduct the corresponding paired t-test analyses (p < 0.05), the presence of significant outliers was checked by visualizing the data using boxplots, the normality of the variables was checked using the Shapiro–Wilk test and by visual inspection using Q–Q plots with 0.95 confidence intervals. Cohen’s d was used to determine the effect size. Graphical visualization of the results was performed using the corresponding bar plots.

Results

The data analysis previously described and conducted to compare the results of the two tests (Test 1 and Test 2, transparent and assistive modes, respectively) yielded the results described below.

Figure 7 shows the boxplots of the MAD (%) for both biceps and triceps and for each of the considered tests. The boxplots show that the MAD in the biceps brachii decreased during exoskeleton-assisted elbow flexion (Test 2) in comparison with the transparent mode (Test 1). Conversely, the MAD in the triceps brachii increased during exoskeleton-assisted elbow flexion (Test 2) in comparison with the transparent mode (Test 1), confirming the initial hypothesis. In addition, the antagonistic roles of the biceps brachii and triceps brachii in elbow flexion and extension movements were clearly shown. No significant outliers are identified.

Figure 7 Boxplots of the Muscle Activity Distribution (MAD).

Boxplots of the Muscle Activity Distribution (MAD, %) for the biceps (above) and triceps (below) in Test 1 (transparent mode) and Test 2 (assistive mode).

The following statistical analysis studies on the extent to which these identified differences between Tests 1 and 2 are statistically significant.

The Shapiro–Wilk tests yielded p = 0.52 (>0.05) and, in addition, the visual inspection of the subsequent Q–Q plots made it possible to affirm that the points were located within the previously stated 0.95 confidence intervals. Therefore, the tested data groups were considered as normally distributed.

Figure 8 shows the bar plots of the mean values ±standard deviations of MAD (%) for both the biceps brachii and triceps brachii in Tests 1 and 2, as well as the results from the corresponding paired t-tests: p = 2.26⋅10−7 <0.001 [***], df = 10. Cohen’s d = 3.78 (>0.8, large effect size).

Table 1 summarizes the results from the comparative analysis between Test 1 and Test 2.

Discussion

It is well known that robotic-based interventions involve multiple variables, which is why it is very important to have reliable metrics not only to efficiently configure each rehabilitation session, but also to assess the real impact of these kinds of treatments in processes such as skill acquisition, generalization of motor skills to functional activities, and retention (persistence of the acquired skills) (Krebs et al., 2012).

In this context, this article describes the design and validation of a 3D-printed exoskeleton for upper limb assistance. Our approach involves replicating the functionality of effective exoskeletons previously described in the literature by using more accessible materials, 3D-printing and elastic bands. To validate our device, we performed an electromyography-based assessment of the assistive performance of the exoskeleton for elbow flexion in healthy children as a proof of concept.

Our results provide observable evidence (Fig. 7) of the change (according to the initial hypothesis) that occurs in the MAD for the biceps and triceps when the exoskeleton assists elbow flexion (Test 2) compared to the transparent mode (Test 1). In fact, the statistical analysis affirms that there are statistically significant differences in the MAD when comparing Tests 1 and 2 for both biceps and triceps (Fig. 8) asthe assistance provided to the arm in Test 2 clearly results in lower levels of muscle activation for the biceps. These results are in line with the ones recently published by Dos Anjos et al. (2022), who studied the changes in the distribution of muscle activity when using a passive trunk exoskeleton using high-density sEMG and conclude that the assistive effect of the exoskeleton decreases the average RMS amplitude, implying a decrement in the percentage of muscle activity of the low back muscles for both static and dynamic tasks.

Therefore, based on these results, it can be stated that (1) the exoskeleton correctly assisted elbow flexion, reducing the effect of gravity and, consequently, enabling more upper limb exercises; and (2) the sEMG and, more specifically, the MAD metric made it possible to determine the configuration in which the exoskeleton is operating and distinguish perfectly between the exoskeleton assisting elbow flexion and the exoskeleton in its transparent mode (without assistance). Therefore, the results lead to the conclusion that MAD can be considered a reliable metric for the validation of the assistive performance of the exoskeleton.

In this context and as the differences between the two studied configurations are remarkable according to our results, the MAD could be studied, in a subsequent analysis, as a metric to quantify different levels of assistance (the multilevel quantification and classification could be based on both the number of elastic bands used in each case and the magnitude of the proposed parameter, the MAD). In addition, this parameter could also be studied to be proposed as an objective metric to evaluate selective motor control (SMC), as the results clearly show that the differences in the activation of the biceps brachii and triceps brachii as agonist muscles in elbow flexion-extension can be quantified. Therefore, it is expected that the co-activation of synergist muscles in this or other movements or functional activities can also be measured and quantified. Previous studies have already demonstrated the reliability of sEMG in extracting relevant parameters related to SMC and spasticity (Choi et al., 2018; van den Noort, Scholtes & Harlaar, 2009).

Raouafi, Raison & Achiche (2020) proposed an upper limb motor function index after principal component analysis of kinematics, electromyography, and inertial measurements, which can detect deviation from the upper limb motor function of a typically developing group of children. Therefore, the proposed metric can play a key role in assessing the actual impact of robot-assisted therapies These results open the door to identifying new markers that can quantify the level of skill acquisition, generalization of motor skills to functional activities, and retention of previously acquired skills.

Figure 8 Bar plots of mean values and standard deviations of the Muscle Activity Distribution (MAD).

Bar plots of mean values ± standard deviations of the Muscle Activity Distribution (MAD, %) for biceps (above) and triceps (below). Statistically significant differences between Test 1 (transparent mode) and Test 2 (assistive mode) for both biceps and triceps (p < 0.001 [***]).

Table 1 Summary of the results from the comparative analysis between Test 1 and Test 2.

	Biceps –MAD (%)	Triceps –MAD (%)	
Test 1
(mean ± standard deviation)	76.63 ± 6.98	23.37 ± 6.98	
Test 2
(mean ± standard deviation)	50.27 ± 6.96	49.73 ± 6.96	
Normality test results: Shapiro–Wilk
(p-value)	0.52 > 0.05	
Paired t-test results
(p-value)	2.26 ⋅ 10−7 < 0.001	
Effect size results
(Cohen’s d)	3.78 (large)	

Finally, both the limitations and the strengths of this study should be considered. Regarding the limitations, it is important to note that our sample is composed of healthy children with relatively homogeneous physical and motor characteristics so the results cannot be directly extrapolated (although they can serve as a reference) to children with different kinds of motor impairments as, in this case, much more heterogeneity in the physical and motor development conditions is expected. In addition, it is important to highlight that although in this proof of concept, a relatively small sample was enough to obtain statistically significant results with a large effect size (again possibly due to the homogeneity of the sample), it is expected to need a larger sample for the validation of the exoskeleton in children with motor diseases. Regarding the strengths, it is important to highlight that this study presents a simple metric (the MAD), that is very easy to obtain (with the suitable electromyography system) and makes it possible to uniquely identify whether or not the device is performing the assistance task, thus allowing a quick and efficient validation of this functionality.

Conclusions

POWERUP is a low-cost and easy-to-use passive exoskeleton for upper limb whose assistive performance (tested for assisting elbow flexion in a sample of healthy children as a proof of concept) can be easily validated through an electromyography-based assessment.

According to the results, there were statistically significant differences in our sample in the Muscle Activity Distribution (MAD, %) for both biceps and triceps between the transparent mode (no assistance provided) and the assistive mode. It can be stated that (1) the exoskeleton correctly assisted elbow flexion, reducing the effect of gravity and (2) the MAD can be considered a reliable metric for the validation of the assistive performance of the exoskeleton. Overall, it can be said that sEMG was a powerful tool for both (1) the assessment of the assistance capacity in elbow flexion of the POWERUP exoskeleton tested in healthy children as a proof of concept and, by extension, of other similar upper limb exoskeletons, and (2) it offers an interesting metric, the MAD, to be more deeply studied for both the evaluation of the SMC and the impact of robot-assisted therapies in children with motor disabilities.

Supplemental Information

Supplemental Information 1 Raw values of the Mean Activity Distribution (MAD)

The value of the Mean Activity Distribution (MAD) metric for each subject for each configuration: transparent mode vs. assistive mode. These values were used for statistical analysis to compare these two modes of operation.

Click here for additional data file.

The authors thank Colegio San Pablo CEU Montepríncipe (Madrid, Spain) for their participation and collaboration in the experiments.

Additional Information and Declarations

Competing Interests

Author Contributions

Human Ethics

Data Availability

The authors declare there are no competing interests.

Cristina Sanchez conceived and designed the experiments, performed the experiments, analyzed the data, prepared figures and/or tables, authored or reviewed drafts of the article, and approved the final draft.

Laura Blanco conceived and designed the experiments, performed the experiments, analyzed the data, prepared figures and/or tables, authored or reviewed drafts of the article, and approved the final draft.

Carmina del Río conceived and designed the experiments, performed the experiments, authored or reviewed drafts of the article, and approved the final draft.

Eloy Urendes analyzed the data, authored or reviewed drafts of the article, and approved the final draft.

Vanina Costa analyzed the data, authored or reviewed drafts of the article, and approved the final draft.

Rafael Raya conceived and designed the experiments, authored or reviewed drafts of the article, and approved the final draft.

The following information was supplied relating to ethical approvals (i.e., approving body and any reference numbers):

All children, their parents or legal guardians, and the head office of Colegio San Pablo CEU Montepríncipe provided consent to participate in the study. Ethical approval was obtained from the Research Ethics Committee of the San Pablo CEU University (561/21/53).

The following information was supplied regarding data availability:

The raw measurements are available in the Supplemental Files.

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
