# Peer review of "A 3D-printed passive exoskeleton for upper limb assistance in children with motor disorders: proof of concept through an electromyography-based assessment"

_PeerJ, doi:10.7717/peerj.15095_

## Round 0.1 · original submission · Major Revisions

Dear authors,

Please respond accurately point by point to the reviewers' comments and resubmit.

·

Basic reporting

• Improve historical background about exoskeleton and the application in clinical fields at line 72-74
• English language should be improved in INTRODUCTION: for example, the phrases between lines 92 and 95 could be better described in order to understand the role of sEMG and the meaning of RMS. Furthermore, line 105 “on the other hand, and,…” and line 109 “…It can be said…” should be better expressed.
• I suggest to better organize the paragraph about EMG in to introduction: improve description and meaning of amplitude parameter (RMS) for static and dynamic task in heathy and neuromuscular diseases and report the benefits of sEMG in the first part of it (not in the end).
• The hypothesizes described between lines 112 and 117 should be better reported ( I suggest to enumerate them).
• In line 100 are reported 2 references “Xiao et al. (22, 23)” but only one of them is described
• In text references have been reported by “number” and not by “author’s name and year of publications”
• Abstract: more information about sample size and results could be report
• References (bibliography) don’t follow PeerJ Guidelines:
o report journal full name and all author’s name - Journal reference format: List of authors (with initials). Publication year. Full article title. Full title of the Journal, volume: page extents. DOI (if available).
o some references (book) are incomplete

Experimental design

• Is there a reference about exoskeleton printing’s method? If yes, it could be interesting to insert it
• Is there a specific reference for your EMG analysis (Muscle activity distribution) in exoskeleton studies?
Check following errors:
• Line 162: correct the figure number
• Line 189: correct the figure number
• Line 197 / 198: the word “project” is reported 2 times in SENIAM phrase
• Line 198: which is the exact position of electrodes placement in fossa cubit? (it should be around 1/3 of it)
• Line 220-221: improve English languages for “against which to compare”
• Line 231: correct English “was to flex”

• Is there a reference for this EMG recording protocol (from line 227 to line 234)?
• Line 239-242: the two purposes should be better described – improve english

Validity of the findings

• Discussion should be implemented with more studies comparison about EMG and types of exoskeleton. Only one research (39) is described for it.
• The recent research of FV Dos Anjos “ Changes in the distribution of muscle activity when using a passive trunk exoskeleton depend on the type of working task: A high-density surface EMG study” Journal of Biomechanics 130 (2022) 110846 – could be helpful
• Conclusions should report that participants are healthy children, so the validation of exoskeleton is related to this sample. Furthermore, the limits of the study must be clearly defined

Additional comments

Generally, english should be reviewed and improved in the sections reported above.

Reviewer 2 ·

Basic reporting

'no comment'

Experimental design

'no comment'

Validity of the findings

'no comment'

Additional comments

The authors addressed an interesting study with a very current topic. however, I have some suggestions for the authors to improve the manuscript.

Abstract
It is written correctly. Gives highlights from each section of the paper.
Introduction
I could suggest to the authors to insert or better describe what kind of pediatric diseases would benefit from this technology. There are some pediatric chronic degenerative diseases that could benefit from this technology also as a re-educational path.

- Patti A, Maggio MC, Corsello G, Messina G, Iovane A, Palma A. Evaluation of Fitness and the Balance Levels of Children with a Diagnosis of Juvenile Idiopathic Arthritis: A Pilot Study. Int J Environ Res Public Health. 2017 Jul 19;14(7):806. doi: 10.3390/ijerph14070806. PMID: 28753965; PMCID: PMC5551244.

- Jeannet PY, Aminian K, Bloetzer C, Najafi B, Paraschiv-Ionescu A. Continuous monitoring and quantification of multiple parameters of daily physical activity in ambulatory Duchenne muscular dystrophy patients. Eur J Paediatr Neurol. 2011 Jan;15(1):40-7. doi: 10.1016/j.ejpn.2010.07.002. Epub 2010 Aug 17. PMID: 20719551.

- Fonvig CE, Troelsen J, Dunkhase-Heinl U, Lauritsen JM, Holsgaard-Larsen A. Predictors of physical activity levels in children and adolescents with cerebral palsy: clinical cohort study protocol. BMJ Open. 2021 Sep 21;11(9):e047522. doi: 10.1136/bmjopen-2020-047522. PMID: 34548350; PMCID: PMC8458314.

Methods
Overall, the methodology is clearly explained.
The tools used are validated and reliable.
The statistical techniques used are appropriate. Participants
-How did you decide on the sample size before starting the study? Have you carried out a priori power analysis? For example, with G*Power
Results
I would like to suggest to the authors to insert a table with the results and the analysis of the comparative data

Discussion
The discussions are clear and to point.
It is necessary to insert the limits of the study

---

## Round 0.2 · Minor Revisions

Dear authors,

You have addressed all reviewer comments appropriately.

The manuscript is almost ready to be published, however, the Section Editor noted a few minor edits and suggestions - please see the attached PDF.

Reviewer 2 ·

Basic reporting

I am satisfied with the current version.

Experimental design

I am satisfied with the current version.

Validity of the findings

I am satisfied with the current version.

Additional comments

I am satisfied with the current version.

---

## Round 0.3 · accepted · Accept

Dear authors,

Congratulations!

You have carefully followed the suggestions of the reviewers and made the appropriate revisions. The manuscript is much improved.